# Surfing: Iterative Optimization Over Incrementally Trained Deep Networks

**Ganlin Song**
Department of Statistics and Data Science
Yale University
ganlin.song@yale.edu

**Zhou Fan**
Department of Statistics and Data Science
Yale University
zhou.fan@yale.edu

**John Lafferty**
Department of Statistics and Data Science
Yale University
john.lafferty@yale.edu

## Abstract

We investigate a sequential optimization procedure to minimize the empirical risk functional $f_{\widehat{\theta}}(x) = \frac{1}{2}\|G_{\widehat{\theta}}(x) - y\|^2$ for certain families of deep networks $G_\theta(x)$. The approach is to optimize a sequence of objective functions that use network parameters obtained during different stages of the training process. When initialized with random parameters $\theta_0$, we show that the objective $f_{\theta_0}(x)$ is "nice" and easy to optimize with gradient descent. As learning is carried out, we obtain a sequence of generative networks $x \mapsto G_{\theta_t}(x)$ and associated risk functions $f_{\theta_t}(x)$, where $t$ indicates a stage of stochastic gradient descent during training. Since the parameters of the network do not change by very much in each step, the surface evolves slowly and can be incrementally optimized. The algorithm is formalized and analyzed for a family of expansive networks. We call the procedure *surfing* since it rides along the peak of the evolving (negative) empirical risk function, starting from a smooth surface at the beginning of learning and ending with a wavy nonconvex surface after learning is complete. Experiments show how surfing can be used to find the global optimum and for compressed sensing even when direct gradient descent on the final learned network fails.

## 1 Introduction

Intensive recent research has provided insight into the performance and mathematical properties of deep neural networks, improving understanding of their strong empirical performance on different types of data. Some of this work has investigated gradient descent algorithms that optimize the weights of deep networks during learning (Du et al., 2018b,a; Davis et al., 2018; Li and Yuan, 2017; Li and Liang, 2018). In this paper we focus on optimization over the inputs to an already trained deep network in order to best approximate a target data point. Specifically, we consider the least squares objective function

$$f_{\widehat{\theta}}(x) = \frac{1}{2}\|G_{\widehat{\theta}}(x) - y\|^2$$

where $G_\theta(x)$ denotes a multi-layer feed-forward network and $\widehat{\theta}$ denotes the parameters of the network after training. The network is considered to be a mapping from a latent input $x \in \mathbb{R}^k$ to an output $G_\theta(x) \in \mathbb{R}^n$ with $k \ll n$. A closely related objective is to minimize $f_{\theta,A}(x) = \frac{1}{2}\|AG_\theta(x) - Ay\|^2$ where $A$ is a random matrix.

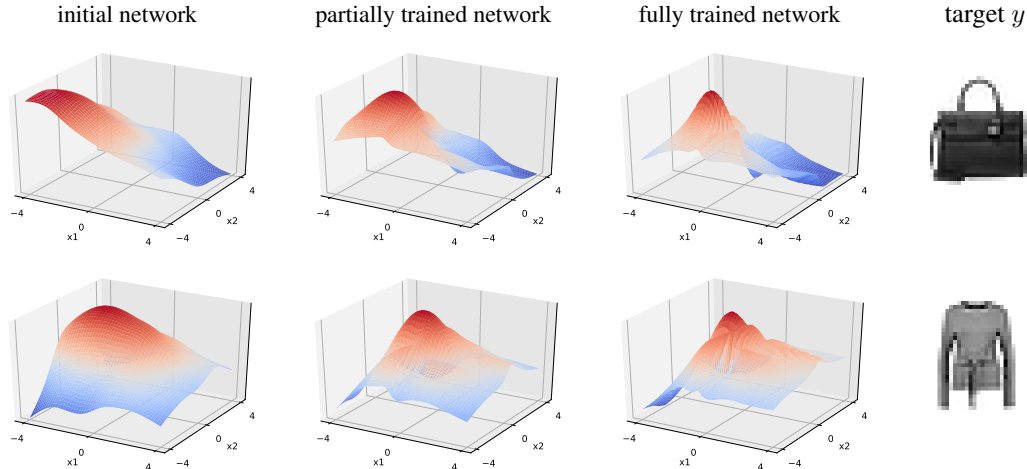

initial network      partially trained network      fully trained network      target $y$

Figure 1: Behavior of the surfaces $x \mapsto -\frac{1}{2}\|G_{\theta_t}(x) - y\|^2$ for two targets $y$ shown for three levels of training, from random networks (left) to fully trained networks (right) on Fashion MNIST data. The network structure has two fully connected layers and two transposed convolution layers with batch normalization, trained as a VAE.

Hand and Voroninski (2019) study the behavior of the function $f_{\theta_0, A}$ in a compressed sensing framework where $y = G_{\theta_0}(x_0)$ is generated from a random network with parameters $\theta_0 = (W_1, \ldots, W_d)$ drawn from Gaussian matrix ensembles; thus, the network is not trained. In this setting, it is shown that the surface is very well behaved. In particular, outside of small neighborhoods around $x_0$ and a scalar multiple of $-x_0$, the function $f_{\theta_0, A}(x)$ always has a descent direction.

When the parameters of the network are trained, the landscape of the function $f_{\widehat{\theta}}(x)$ can be complicated; it will in general be nonconvex with multiple local optima. Figure 1 illustrates the behavior of the surfaces as they evolve from random networks (left) to fully trained networks (right) for 4-layer networks trained on Fashion MNIST using a variational autoencoder. For each of two target values $y$, three surfaces $x \mapsto -\frac{1}{2}\|G_{\theta_t}(x) - y\|^2$ are shown for different levels of training.

This paper explores the following simple idea. We incrementally optimize a sequence of objective functions $f_{\theta_0}, f_{\theta_1}, \ldots, f_{\theta_T}$ where the parameters $\theta_0, \theta_1, \ldots, \theta_T = \widehat{\theta}$ are obtained using stochastic gradient descent in $\theta$ during training. When initialized with random parameters $\theta_0$, we show that the empirical risk function $f_{\theta_0}(x) = \frac{1}{2}\|G_{\theta_0}(x) - y\|^2$ is "nice" and easy to optimize with gradient descent. As learning is carried out, we obtain a sequence of generative networks $x \mapsto G_{\theta_t}(x)$ and associated risk functions $f_{\theta_t}(x)$, where $t$ indicates an intermediate stage of stochastic gradient descent during training. Since the parameters of the network do not change by very much in each step (Du et al., 2018a,b), the surface evolves slowly. We initialize $x$ for the current network $G_{\theta_t}(x)$ at the optimum $x_{t-1}^*$ found for the previous network $G_{\theta_{t-1}}(x)$ and then carry out gradient descent to obtain the updated point $x_t^* = \operatorname{argmin}_x f_{\theta_t}(x)$.

We call this process *surfing* since it rides along the peaks of the evolving (negative) empirical risk function, starting from a smooth surface at the beginning of learning and ending with a wavy nonconvex surface after learning is complete. We formalize this algorithm in a manner that makes it amenable to analysis. First, when $\theta_0$ is initialized so that the weights are random Gaussian matrices, we prove a theorem showing that the surface has a descent direction at each point outside of a small neighborhood. The analysis of Hand and Voroninski (2019) does not directly apply in our case since the target $y$ is an arbitrary test point, and not necessarily generated according to the random network. We then give an analysis that describes how projected gradient descent can be used to proceed from the optimum of one network to the next. Our approach is based on the fact that the ReLU network and squared error objective result in a piecewise quadratic surface. Experiments are run to show how surfing can be used to find the global optimum and for compressed sensing even when direct gradient descent fails, using several experimental setups with networks trained with both VAE and GAN techniques.

## 2   Background and Previous Results

In this work we treat the problem of approximating an observed vector $y$ in terms of the output $G_{\widehat{\theta}}(x)$ of a trained generative model. Traditional generative processes such as graphical models are statistical models that define a distribution over a sample space. When deep networks are viewed as generative models, the distribution is typically singular, being a deterministic mapping of a low-dimensional latent random vector to a high-dimensional output space. Certain forms of "reversible deep networks" allow for the computation of densities and inversion (Dinh et al., 2017; Kingma and Dhariwal, 2018; Chen et al., 2018).

The variational autoencoder (VAE) approach training a generative (decoder) network is to model the conditional probability of $x$ given $y$ as Gaussian with mean $\mu(y)$ and covariance $\Sigma(y)$ assuming that *a priori* $x \sim N(0, I_k)$ is Gaussian. The mean and covariance are treated as the output of a secondary (encoder) neural network. The two networks are trained by maximizing the evidence lower bound (ELBO) with coupled gradient descent algorithms—one for the encoder network, the other for the decoder network $G_\theta(x)$ (Kingma and Welling, 2014). Whether fitting the networks using a variational or GAN approach (Goodfellow et al., 2014; Arjovsky et al., 2017), the problem of "inverting" the network to obtain $x^* = \operatorname{argmin} f_\theta(x)$ is not addressed by the training procedure.

In the now classical compressed sensing framework (Candes et al., 2006; Donoho et al., 2006), the problem is to reconstruct a sparse signal after observing multiple linear measurements, possibly with added noise. More recent work has begun to investigate generative deep networks as a replacement for sparsity in compressed sensing. Bora et al. (2017) consider identifying $y = G(x_0)$ from linear measurements $Ay$ by optimizing $f(x) = \frac{1}{2}\|Ay - AG(x)\|^2$. Since this objective is nonconvex, it is not guaranteed that gradient descent will converge to the true global minimum. However, for certain classes of ReLU networks it is shown that so long as a point $\widehat{x}$ is found for which $f(\widehat{x})$ is sufficiently close to zero, then $\|y - G(\widehat{x})\|$ is also small. For the case where $y$ does not lie in the image of $G$, an oracle type bound is shown implying that the solution $\widehat{x}$ satisfies $\|G(\widehat{x}) - y\|^2 \leq C \inf_x \|G(x) - y\|^2 + \delta$ for some small error term $\delta$. The authors observe that in experiments the error seems to converge to zero when $\widehat{x}$ is computed using simple gradient descent; but an analysis of this phenomenon is not provided.

Hand and Voroninski (2019) establish the important result that for a $d$-layer random network and random measurement matrix $A$, the least squares objective has favorable geometry, meaning that outside two small neighborhoods there are no first order stationary points, neither local minima nor saddle points. We describe their setup and result in some detail, since it provides a springboard for the surfing algorithm.

Let $G : \mathbb{R}^k \to \mathbb{R}^n$ be a $d$-layer fully connected feedforward generative neural network, which has the form

$$G(x) = \sigma(W_d...\sigma(W_2\sigma(W_1 x))...)$$

where $\sigma$ is the ReLU activation function. The matrix $W_i \in R^{n_i \times n_{i-1}}$ is the set of weights for the $i$th layer and $n_i$ is number of the neurons in this layer with $k = n_0 < n_1 < ... < n_d = n$. If $x_0 \in \mathbb{R}^k$ is the input then $AG(x_0)$ is a set of random linear measurements of the signal $y = G(x_0)$. The objective is to minimize $f_{A,\theta_0}(x) = \frac{1}{2}\left\|AG_{\theta_0}(x) - AG_{\theta_0}(x_0)\right\|^2$ where $\theta_0 = (W_1, \ldots, W_d)$ is the set of weights.

Due to the fact that the nonlinearities $\sigma$ are rectified linear units, $G_{\theta_0}(x)$ is a piecewise linear function. It is convenient to introduce notation that absorbs the activation $\sigma$ into weight matrix $W_i$, denoting

$$W_{+,x} = \operatorname{diag}(Wx > 0)W.$$

For a fixed $W$, the matrix $W_{+,x}$ zeros out the rows of $W$ that do not have a positive dot product with $x$; thus, $\sigma(Wx) = W_{+,x}x$. We further define $W_{1,+,x} = \operatorname{diag}(W_1 x > 0) W_1$ and

$$W_{i,+,x} = \operatorname{diag}(W_i W_{i-1,+,x}...W_{1,+,x}x > 0) W_i.$$

With this notation, we can rewrite the generative network $G_{\theta_0}$ in what looks like a linear form,

$$G_{\theta_0}(x) = W_{d,+,x}W_{d-1,+,x}...W_{1,+,x}x,$$

noting that each matrix $W_{i,+,x}$ depends on the input $x$.

If $f_{A,\theta_0}(x)$ is differentiable at $x$, we can write the gradient as

$$\nabla f_{A,\theta_0}(x) = \Big(\prod_{i=d}^{1} W_{i,+,x}\Big)^T A^T A\Big(\prod_{i=d}^{1} W_{i,+,x}\Big)x - \Big(\prod_{i=d}^{1} W_{i,+,x}\Big)^T A^T A\Big(\prod_{i=d}^{1} W_{i,+,x_0}\Big)x_0.$$

In this expression, one can see intuitively that under the assumption that $A$ and $W_i$ are Gaussian matrices, the gradient $\nabla f_{\theta_0}(x)$ should concentrate around a deterministic vector $v_{x,x_0}$. Hand and Voroninski (2019) establish sufficient conditions for concentration of the random matrices around deterministic quantities, so that $v_{x,x_0}$ has norm bounded away from zero if $x$ is sufficiently far from $x_0$ or a scalar multiple of $-x_0$. Their results show that for random networks having a sufficiently expansive number of neurons in each layer, the objective $f_{A,\theta_0}$ has a landscape favorable to gradient descent.

We build on these ideas, showing first that optimizing with respect to $x$ for a random network and arbitrary signal $y$ can be done with gradient descent. This requires modified proof techniques, since it is no longer assumed that $y = G_{\theta_0}(x_0)$. In fact, $y$ can be arbitrary and we wish to approximate it as $G_{\widehat{\theta}}(x(y))$ for some $x(y)$. Second, after this initial optimization is carried out, we show how projected gradient descent can be used to track the optimum as the network undergoes a series of small changes. Our results are stated formally in the following section.

## 3  Theoretical Results

Suppose we have a sequence of networks $G_0, G_1, \ldots, G_T$ generated from the training process. For instance, we may take a network with randomly initialized weights as $G_0$, and record the network after each step of gradient descent in training; $G_T = G$ is the final trained network.

For a given vector $y \in \mathbb{R}^n$, we wish to minimize the objective $f(x) = \frac{1}{2}\|AG(x) - Ay\|^2$ with respect to $x$ for the final network $G$, where either $A = I \in \mathbb{R}^{n \times n}$, or $A \in \mathbb{R}^{m \times n}$ is a measurement matrix with i.i.d. $\mathcal{N}(0, 1/m)$ entries in a compressed sensing context. Write

$$f_t(x) = \frac{1}{2}\|AG_t(x) - Ay\|^2, \quad \forall\, t \in [T]. \quad (1)$$

The idea is that we first minimize $f_0$, which has a nicer landscape, to obtain the minimizer $x_0$. We then apply gradient descent on $f_t$ for $t = 1, 2, ..., T$ successively, starting from the minimizer $x_{t-1}$ for the previous network.

---
**Algorithm 1** Surfing
---
**Input:** Sequence of networks $\theta_0, \theta_1, \ldots, \theta_T$
1: $x_{-1} \leftarrow 0$
2: **for** $t = 0$ to $T$ **do**
3:     $x \leftarrow x_{t-1}$
4:     **repeat**
5:         $x \leftarrow x - \eta \nabla f_{\theta_t}(x)$
6:     **until** convergence
7:     $x_t \leftarrow x$
**Output:** $x_T$

---

We provide some theoretical analysis in partial support of this algorithmic idea. First, we show that at random initialization $G_0$, all critical points of $f_0(x)$ are localized to a small ball around zero. Second, we show that if $G_0, \ldots, G_T$ are obtained from a discretization of a continuous flow, along which the global minimizer of $f_t(x)$ is unique and Lipschitz-continuous, then a projected-gradient version of surfing can successively find the minimizers for $G_1, \ldots, G_T$ starting from the minimizer for $G_0$.

We consider expansive feedforward neural networks $G : \mathbb{R}^k \times \Theta \mapsto \mathbb{R}^n$ given by

$$G(x, \theta) = V\sigma(W_d \ldots \sigma(W_2\sigma(W_1 x + b_1) + b_2)\ldots + b_d).$$

Here, $d$ is the number of intermediate layers (which we will treat as constant), $\sigma$ is the ReLU activation function $\sigma(x) = \max(x, 0)$ applied entrywise, and $\theta = (V, W_1, ..., W_d, b_1, ..., b_d)$ are the network parameters. The input dimension is $k \equiv n_0$, each intermediate layer $i \in [d]$ has weights $W_i \in \mathbb{R}^{n_i \times n_{i-1}}$ and biases $b_i \in \mathbb{R}^{n_i}$, and a linear transform $V \in \mathbb{R}^{n \times n_d}$ is applied in the final layer.

For our first result, consider fixed $y \in \mathbb{R}^n$ and a random initialization $G_0(x) \equiv G(x, \theta_0)$ where $\theta_0$ has Gaussian entries (independent of $y$). If the network is sufficiently expansive at each intermediate layer, then the following shows that with high probability, all critical points of $f_0(x)$ belong to a small ball around 0. More concretely, the directional derivative $D_{-x/\|x\|}f_0(x)$ satisfies

$$D_{-x/\|x\|}f_0(x) \equiv \lim_{t \to 0^+} \frac{f_0(x - tx/\|x\|) - f_0(x)}{t} < 0. \quad (2)$$

Thus $-x/\|x\|$ is a first-order descent direction of the objective $f_0$ at $x$.

**Theorem 3.1.** *Fix $y \in \mathbb{R}^n$. Let $V$ have $\mathcal{N}(0, 1/n)$ entries, let $b_i$ and $W_i$ have $\mathcal{N}(0, 1/n_i)$ entries for each $i \in [d]$, and suppose these are independent. There exist $d$-dependent constants $C, C', c, \varepsilon_0 > 0$ such that for any $\varepsilon \in (0, \varepsilon_0)$, if*

1. *$n \geq n_d$ and $n_i > C(\varepsilon^{-2} \log \varepsilon^{-1}) n_{i-1} \log n_i$ for all $i \in [d]$, and*

2. *Either $A = I$ and $m = n$, or $A \in \mathbb{R}^{m \times n}$ has i.i.d. $\mathcal{N}(0, 1/m)$ entries (independent of $V, \{b_i\}, \{W_i\}$) where $m \geq Ck(\varepsilon^{-1} \log \varepsilon^{-1}) \log(n_1 \ldots n_d)$,*

*then with probability at least $1 - C(e^{-c\varepsilon m} + n_d e^{-c\varepsilon^4 n_{d-1}} + \sum_{i=1}^{d-1} n_i e^{-c\varepsilon^2 n_{i-1}})$, every $x \in \mathbb{R}^k$ outside the ball $\|x\| \leq C'\varepsilon(1 + \|y\|)$ satisfies (2).*

We defer the proof to the supplementary material. Note that if instead $G_0$ were correlated with $y$, say $y = G_0(x_*)$ for some input $x_*$ with $\|x_*\| \asymp 1$, then $x_*$ would be a global minimizer of $f_0(x)$, and we would have $\|y\| \asymp \|x_d\| \asymp \ldots \asymp \|x_1\| \asymp \|x_*\| \asymp 1$ in the above network where $x_i \in \mathbb{R}^{n_i}$ is the output of the $i^{\text{th}}$ layer. The theorem shows that for a random initialization of $G_0$ which is independent of $y$, the minimizer is instead localized to a ball around 0 which is smaller in radius by the factor $\varepsilon$.

For our second result, consider a network flow

$$G^s(x) \equiv G(x, \theta(s))$$

for $s \in [0, S]$, where $\theta(s) = (V(s), W_1(s), b_1(s), \ldots, W_d(s), b_d(s))$ evolve continuously in a time parameter $s$. As a model for network training, we assume that $G_0, \ldots, G_T$ are obtained by discrete sampling from this flow via $G_t = G^{\delta t}$, corresponding to $s \equiv \delta t$ for a small time discretization step $\delta$.

We assume boundedness of the weights and uniqueness and Lipschitz-continuity of the global minimizer along this flow.

**Assumption 3.2.** *There are constants $M, L < \infty$ such that*

1. *For every $i \in [d]$ and $s \in [0, S]$,*
$$\|W_i(s)\| \leq M.$$

2. *The global minimizer $x_*(s) = \operatorname{argmin}_x f(x, \theta(s))$ is unique and satisfies*
$$\|x_*(s) - x_*(s')\| \leq L|s - s'|$$
*where $f(x, \theta(s)) = \frac{1}{2}\|AG(x, \theta(s)) - Ay\|^2$.*

Fixing $\theta$, the function $G(x, \theta)$ is continuous and piecewise-linear in $x$. For each $x \in \mathbb{R}^k$, there is at least one linear piece $P_0$ (a polytope in $\mathbb{R}^k$) of this function that contains $x$. For a slack parameter $\tau > 0$, consider the rows given by

$$S(x, \theta, \tau) = \{(i, j) : |w_{i,j}^\top x_{i-1} + b_{i,j}| \leq \tau\},$$

where

$$x_{i-1} = \sigma(W_{i-1} \ldots \sigma(W_1 x + b_1) \ldots + b_{i-1})$$

is the output of the $(i-1)^{\text{th}}$ layer for this input $x$, and $v_j^\top$, $w_{i,j}^\top$, and $b_{i,j}$ are respectively the $j^{\text{th}}$ row of $V$, the $j^{\text{th}}$ row of $W_i$ and the $j^{\text{th}}$ entry of $b_i$ in $\theta$. This set $S(x, \theta, \tau)$ represents those neurons that are close to 0 before ReLU thresholding, and hence whose activations may change after a small change of the network input $x$. Define

$$\mathcal{P}(x, \theta, \tau) = \{P_0, P_1, \ldots, P_G\}$$

as the set of all linear pieces $P_g$ whose activation patterns differ from $P_0$ only in rows belonging to $S(x, \theta, \tau)$. That is, for every $x' \in P_g \in \mathcal{P}(x, \theta, \tau)$ and $(i, j) \notin S(x, \theta, \tau)$, we have

$$\operatorname{sign}(w_{i,j}^\top x_{i-1}' + b_{i,j}) = \operatorname{sign}(w_{i,j}^\top x_{i-1} + b_{i,j})$$

where $x_{i-1}'$ is the output of the $(i-1)^{\text{th}}$ layer for input $x'$.

With this definition, we consider a stylized projected-gradient surfing procedure in Algorithm 2, where $\operatorname{Proj}_P$ is the orthogonal projection onto the polytope $P$.

---

**Algorithm 2** Projected-gradient Surfing

---

**Input:** Network flow $\{G(\cdot, \theta(s)) : s \in [0, S]\}$, parameters $\delta, \tau, \eta > 0$.
1: Initialize $x_0 = \operatorname{argmin}_x f(x, \theta(0))$.
2: **for** $t = 1, \ldots, T$ **do**
3:     **for** each linear piece $P_g \in \mathcal{P}(x_{t-1}, \theta(\delta t), \tau)$ **do**
4:         $x \leftarrow x_{t-1}$
5:         **repeat**
6:             $x \leftarrow \operatorname{Proj}_{P_g}(x - \eta \nabla f(x, \theta(\delta t)))$
7:         **until** convergence
8:         $x_t^{(g)} \leftarrow x$
9:     $x_t \leftarrow x_t^{(g)}$ for $g \in \{0, \ldots, G\}$ that achieves the minimum value of $f(x_t^{(g)}, \theta(\delta t))$.
**Output:** $x_T$

---

The complexity of this algorithm depends on the number of pieces $G$ to be optimized over in each step. We expect this to be small in practice when the slack parameter $\tau$ is chosen sufficiently small, and provide a heuristic argument in the supplement indicating why this may be the case.

The following shows that for any $\tau > 0$, there is a sufficiently fine time discretization $\delta$ depending on $\tau, M, L$ such that Algorithm 2 tracks the global minimizer. In particular, for the final objective $f_T(x) = f(x, \theta(\delta T))$ corresponding to the network $G_T$, the output $x_T$ is the global minimizer of $f_T(x)$. We remark that the time discretization $\delta$ may need to be smaller for deeper networks, as $G(x)$ corresponding to a deeper network may have a larger Lipschitz constant in $x$. The specific dependence below arises from bounding this Lipschitz constant by $\prod_{i=1}^{d} \|W_i\|$, which is a conservative bound also used and discussed in greater detail in Szegedy et al. (2014); Virmaux and Scaman (2018).

**Theorem 3.3.** *Suppose Assumption 3.2 holds. For any $\tau > 0$, if $\delta < \tau/(L \max(M, 1)^{d+1})$ and $x_0 = \operatorname{argmin}_x f(x, \theta(0))$, then the iterates $x_t$ in Algorithm 2 are given by $x_t = \operatorname{argmin}_x f(x, \theta(\delta t))$ for each $t = 1, \ldots, T$.*

*Proof.* For any fixed $\theta$, let $x, x' \in \mathbb{R}^k$ be two inputs to $G(x, \theta)$. If $x_i, x_i'$ are the corresponding outputs of the $i^{\text{th}}$ layer, using the assumption $\|W_i\| \leq M$ and the fact that the ReLU activation $\sigma$ is 1-Lipschitz, we have

$$\begin{aligned}
\|x_i - x_i'\| &= \|\sigma(W_i x_{i-1} + b_i) - \sigma(W_i x_{i-1}' + b_i)\| \\
&\leq \|(W_i x_{i-1} + b_i) - (W_i x_{i-1}' + b_i)\| \\
&\leq M\|x_{i-1} - x_{i-1}'\| \leq \ldots \leq M^i\|x - x'\|.
\end{aligned}$$

Let $x_*(s) = \operatorname{argmin}_x f(x, \theta(s))$. By assumption, $\|x_*(s - \delta) - x_*(s)\| \leq L\delta$. For the network with parameter $\theta(s)$ at time $s$, let $x_{*,i}(s)$ and $x_{*,i}(s - \delta)$ be the outputs at the $i^{\text{th}}$ layer corresponding to inputs $x_*(s)$ and $x_*(s - \delta)$. Then for any $i \in [d]$ and $j \in [n_i]$, the above yields

$$|(w_{i,j}(s)^\top x_{*,i}(s - \delta) + b_{i,j}) - (w_{i,j}(s)^\top x_{*,i}(s) + b_{i,j})| \leq \|w_{i,j}(s)\|\|x_{*,i}(s - \delta) - x_{*,i}(s)\|$$
$$\leq M \cdot M^i \|x_*(s - \delta) - x_*(s)\| \leq M^{i+1} L\delta.$$

For $\delta < \tau/(L \max(M, 1)^{d+1})$, this implies that for every $(i, j)$ where $|w_{i,j}(s)^\top x_{*,i}(s-\delta) + b_{i,j}| \geq \tau$, we have

$$\operatorname{sign}(w_{i,j}(s)^\top x_{*,i}(s - \delta) + b_{i,j}) = \operatorname{sign}(w_{i,j}(s)^\top x_{*,i}(s) + b_{i,j}).$$

That is, $x_*(s) \in P_g$ for some $P_g \in \mathcal{P}(x_*(s - \delta), \theta(s), \tau)$.

Assuming that $x_{t-1} = x_*(\delta(t - 1))$, this implies that the next global minimizer $x_*(\delta t)$ belongs to some $P_g \in \mathcal{P}(x_{t-1}, \theta(\delta t), \tau)$. Since $f(x, \theta(\delta t))$ is quadratic on $P_g$, projected gradient descent over $P_g$ in Algorithm 2 converges to $x_*(\delta t)$, and hence Algorithm 2 yields $x_t = x_*(\delta t)$. The result then follows from induction on $t$. $\qquad\square$

## 4 Experiments

We present experiments to illustrate the performance of surfing over a sequence of networks during training compared with gradient descent over the final trained network. We mainly use the Fashion-

| Input dimension | | 5 | 10 | 20 | 5 | 10 | 20 |
|---|---|---|---|---|---|---|---|
| | Model | | VAE | | | DCGAN | |
| % successful | Regular Adam | 98.7 | 100 | 100 | 48.3 | 68.7 | 80.0 |
| | Surfing | 100 | 100 | 100 | 78.3 | 98.7 | 96.3 |
| # iterations | Regular Adam | 737 | 1330 | 8215 | 618 | 4560 | 18937 |
| | Surfing | 775 | 1404 | 10744 | 741 | 6514 | 33294 |
| | Model | | WGAN | | | WGAN-GP | |
| % successful | Regular Adam | 56.0 | 84.3 | 90.3 | 47.0 | 64.7 | 64.7 |
| | Surfing | 81.7 | 97.3 | 99.3 | 83.7 | 95.7 | 97.3 |
| # iterations | Regular Adam | 464 | 1227 | 3702 | 463 | 1915 | 15445 |
| | Surfing | 547 | 1450 | 4986 | 564 | 2394 | 25991 |

Table 1: Surfing compared against direct gradient descent over the final trained network, for various generative models with input dimensions $k = 5, 10, 20$. Shown are percentages of "successful" solutions $\widehat{x}_T$ satisfying $\|\widehat{x}_T - x_*\| < 0.01$, and 75th-percentiles of the total number of gradient descent steps used (across all networks $G_0, \ldots, G_T$ for surfing) until $\|\widehat{x}_T - x_*\| < 0.01$ was reached.

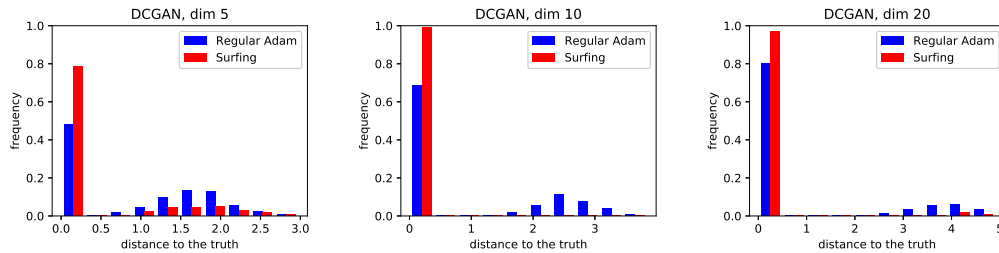

Figure 2: Distribution of distance between solution $\widehat{x}_T$ and the truth $x_*$ for DCGAN trained models, comparing surfing (red) to regular gradient descent (blue) over the final network. Both procedures use Adam in their gradient descent computations. The results indicate that direct descent often succeeds, but can also converge to a point that is far from the optimum. By moving along the optimum of the evolving surface, surfing is able to move closer to the optimum in these cases.

MNIST dataset to carry out the simulations, which is similar to MNIST in many characteristics, but is more difficult to train. We build multiple generative models, trained using VAE (Kingma and Welling, 2014), DCGAN (Radford et al., 2015), WGAN (Arjovsky et al., 2017) and WGAN-GP (Gulrajani et al., 2017). The structure of the generator/decoder networks that we use are the same as those reported by Chen et al. (2016); they include two fully connected layers and two transposed convolution layers with batch normalization after each layer (Ioffe and Szegedy, 2015). We use the simple surfing algorithm in these experiments, rather than the projected-gradient algorithm proposed for theoretical analysis. Note also that the network architectures do not precisely match the expansive relu networks used in our analysis. Instead, we experiment with architectures and training procedures that are meant to better reflect the current state of the art.

We first consider the problem of minimizing the objective $f(x) = \frac{1}{2}\|G(x) - G(x_*)\|^2$ and recovering the image generated from a trained network $G(x) = G_{\theta_T}(x)$ with input $x_*$. We run surfing by taking a sequence of parameters $\theta_0, \theta_1, ..., \theta_T$ for $T = 100$, where $\theta_0$ are the initial random parameters and the intermediate $\theta_t$'s are taken every 40 training steps, and we use Adam (Kingma and Ba, 2014) to carry out gradient descent in $x$ over each network $G_{\theta_t}$. We compare this to "regular Adam", which uses Adam to optimize over $x$ in only the final trained network $G_{\theta_T}$ for $T = 100$.

To ensure that the runtime of surfing is comparable to that of a single initialization of regular Adam, we do not run Adam until convergence for each intermediate network in surfing. Instead, we use a fixed schedule of iterations for the networks $G_{\theta_0}, \ldots, G_{\theta_{T-1}}$, and run Adam to convergence in only the final network $G_{\theta_T}$. The total number of iterations for networks $G_{\theta_0}, \ldots, G_{\theta_{T-1}}$ is set as the 75th-percentile of the iteration count required for convergence of regular Adam. These are split across the networks proportional to a deterministic schedule that allots more steps to the earlier networks where the landscape of $G(x)$ changes more rapidly, and fewer steps to later networks where this landscape stabilizes.

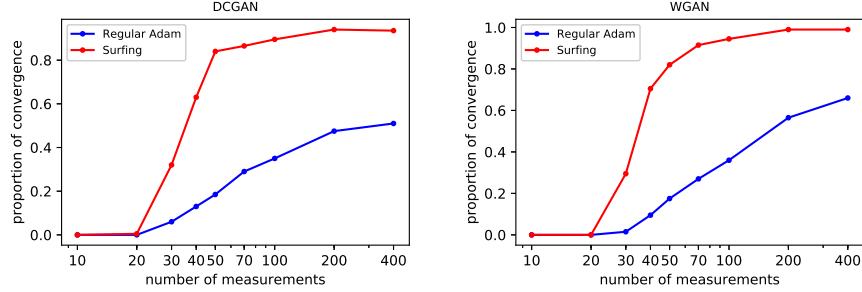

Figure 3: Compressed sensing setting for exact recovery. As a function of the number of random measurements $m$, the lines show the proportion of times surfing (red) and regular gradient descent with Adam (blue) are able to recover the true signal $y = G(x)$, using DCGAN and WGAN.

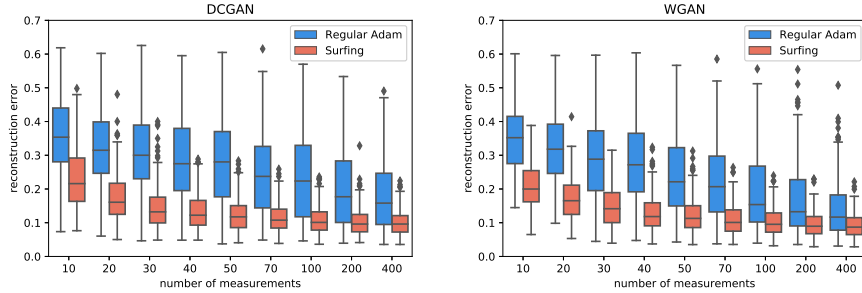

Figure 4: Compressed sensing setting for approximation, or rate-distortion. As a function of the number of random measurements $m$, the box plots summarize the distribution of the per-pixel reconstruction errors for DCGAN and WGAN trained models, using surfing (red) and regular gradient descent with Adam (blue).

For each network training condition, we apply surfing and regular Adam for 300 trials, where in each trial a randomly generated $x_*$ and initial point $x_{init}$ are chosen uniformly from the hypercube $[-1, 1]^k$. Table 1 shows the percentage of trials where the solutions $\widehat{x}_T$ satisfy our criterion for successful recovery $\|\widehat{x}_T - x_*\| < 0.01$, for different models and over three different input dimensions $k$. The table also shows the 75th-percentile for the total number of gradient descent iterations taken (across all networks for surfing), verifying that the runtime of surfing was typically 1–2x that of regular Adam. We also provide the distributions of $\|\widehat{x}_T - x_*\|$ under each setting: Figure 2 shows the results for DCGAN, and results for the other models are collected in the supplementary material.

We next consider the compressed sensing problem with objective $f(x) = \frac{1}{2}\|AG(x) - AG(x_*)\|^2$ where $A \in \mathbb{R}^{m \times n}$ is the Gaussian measurement matrix. We carry out 200 trials for each choice of number of measurements $m$. The parameters $\theta_t$ for surfing are taken every 100 training steps. As before, we record the proportion of the solutions that are close to the truth $x_*$ according to $\|\widehat{x}_T - x_*\| < 0.01$. Figure 3 shows the results for DCGAN and WGAN trained networks with input dimension $k = 20$.

Lastly, we consider the objective $f(x) = \frac{1}{2}\|AG(x) - Ay\|^2$, where $y$ is a real image from the hold-out test data. This can be thought of as a rate-distortion setting, where the error varies as a function of the number of measurements used. We carry out the same experiments as before and compute the average per-pixel reconstruction error $\sqrt{\frac{1}{n}\|G(\widehat{x}_T) - y\|^2}$ as in Bora et al. (2017). Figure 4 shows the distributions of the reconstruction error as the number of measurements $m$ varies.

## 5 Discussion

This paper has explored the idea of incrementally optimizing a sequence of objective risk functions obtained for models that are slowly changing during stochastic gradient descent during training. When initialized with random parameters $\theta_0$, we have shown that the empirical risk function $f_{\theta_0}(x) =$

$\frac{1}{2}\|G_{\theta_0}(x) - y\|^2$ is well behaved and easy to optimize. The surfing algorithm initializes $x$ for the current network $G_{\theta_t}(x)$ at the optimum $x_{t-1}^*$ found for the previous network $G_{\theta_{t-1}}(x)$ and then carries out gradient descent to obtain the updated point $x_t^* = \mathrm{argmin}_x f_{\theta_t}(x)$. Our experiments show that this scheme has merit, and often significantly outperforms direct gradient descent on the final model alone.

On the theoretical side, our main technical result applies and extends ideas of Hand and Voroninski (2019) to show that for random ReLU networks that are sufficiently expansive, the surface of $f_{\theta_0}(x)$ is well-behaved for arbitrary target vectors $y$. This result may be of independent interest, but it is essential for the surfing algorithm because initially the model is poor, with high approximation error. The analysis for the incremental scheme uses projected gradient descent, although we find that simple gradient descent works well in practice. The analysis assumes that the argmin over the surface evolves continuously in training. This assumption is necessary—if the global minimum is discontinuous as a function of $t$, so that the minimizer "jumps" to a far away point, then the surfing procedure will fail in practice.

In our experiments, we see that simple surfing can indeed be effective for mapping outputs $y$ to inputs $x$ for the trained network, where it often outperforms direct gradient descent for a range of deep network architectures and training procedures. However, these simulations also point to the fact that in some settings, direct gradient descent itself can be surprisingly effective. A deeper understanding of this phenomenon could lead to more advanced surfing algorithms that are able to ride to the final optimum even more efficiently and often.

### Acknowledgments

Research supported in part by NSF grants DMS-1513594, CCF-1839308, DMS-1916198, and a J.P. Morgan Faculty Research Award.

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
