[Supplementary Material]

# A  Proof of Theorem 3.1

We denote $[n] = \{1, 2, ..., n\}$, $\Pi_{i=1}^d W_i = W_1 W_2 \ldots W_d$, and $\Pi_{i=d}^1 W_i = W_d W_{d-1} \cdots W_1$. $\|x\|$ and $\|A\|$ are the Euclidean vector norm and matrix operator norm. $C, C', c, c' > 0$ denote $d$-dependent constants that may change from instance to instance.

We adapt ideas of Hand and Voroninski (2017). Denote for simplicity $G(x) = G(x, \theta_0)$ and $f(x) = f_0(x)$. Define

$$W_{i,+,v} = \mathrm{diag}(W_i v + b_i > 0)W_i, \qquad b_{i,+,v} = \mathrm{diag}(W_i v + b_i > 0)b_i$$

where $\mathrm{diag}(w > 0)$ denotes a diagonal matrix with $j$th diagonal element $\mathbb{1}\{w_j > 0\}$. Then

$$\sigma(W_i v + b_i) = W_{i,+,v} v + b_{i,+,v}.$$

The analysis of Hand and Voroninski (2017) shows that the matrices

$$\tilde{W}_{i,+,v} \equiv (W_{i,+,v} \quad b_{i,+,v}) \in \mathbb{R}^{n_i \times (n_{i-1}+1)}$$

satisfy a certain Weight Distribution Condition (WDC), yielding a deterministic approximation for $\tilde{W}_{i,+,v}^\top \tilde{W}_{i,+,v'}$ and any $v, v' \in \mathbb{R}^{n_{i-1}}$. We will use the following consequence of this condition.

**Lemma A.1.** *Under the conditions of Theorem 3.1, with probability at least $1 - C \sum_{i=1}^d n_i e^{-c\varepsilon^2 n_{i-1}}$, the following hold for every $i \in [d]$ and $v, v' \in \mathbb{R}^{n_{i-1}}$:*

*(a) $\|W_{i,+,v}\| \leq 2$ and $\|b_{i,+,v}\| \leq 2$.*

*(b) $\|W_{i,+,v}^\top W_{i,+,v'} - \frac{1}{2}I\| \leq \varepsilon + \theta/\pi$, where $\theta$ is the angle formed by $v$ and $v'$.*

*(c) $\|W_{i,+,v}^\top b_{i,+,v}\| \leq \varepsilon$.*

*Proof.* For (a), note that $\|W_i\| \leq 2$ and $\|b_i\| \leq 2$ with probability $1 - e^{-cn_i}$, by a standard $\chi^2$ tail-bound and operator norm bound for a Gaussian matrix. On the event that these hold, the bounds hold also for $W_{i,+,v}$ and $b_{i,+,v}$ and every $v \in \mathbb{R}^{n_{i-1}}$.

For (b) and (c), by (Hand and Voroninski, 2017, Lemma 11), with probability $1 - 8n_i e^{-c\varepsilon^2 n_{i-1}}$ the matrix $\tilde{W}_{i,+,v}$ satisfies WDC with constant $\varepsilon$ for every $v$. (The dependence of the constants $c, \gamma$ in (Hand and Voroninski, 2017, Lemma 11) are given by $c \gtrsim \varepsilon^{-2} \log \varepsilon^{-1}$ and $\gamma \lesssim \varepsilon^2$ as indicated in the proof. This condition for $c$ matches the growth rate of $n_i$ specified in our Theorem 3.1.) From the form of $Q$ in (Hand and Voroninski, 2017, Definition 2), the WDC implies

$$\left\| \tilde{W}_{i,+,v}^\top \tilde{W}_{i,+,v'} - \frac{1}{2}I \right\| \leq \varepsilon + \tilde{\theta}/\pi$$

where $\tilde{\theta}$ is the angle between $(v, 1)$ and $(v', 1)$. Noting that $\tilde{\theta} \leq \theta$ and recalling the definition of $\tilde{W}_{i,+,v}$, we get (b) and (c). $\qquad\square$

For $x \in \mathbb{R}^k$, let $x_0 = x$ and let $x_i = \sigma(W_i \ldots \sigma(W_1 x + b_1) \ldots + b_i)$ be the output of the $i$th layer. Denote

$$W_{i,x} = W_{i,+,x_{i-1}}, \qquad b_{i,x} = b_{i,+,x_{i-1}}.$$

Then also $x_i = W_{i,x} x_{i-1} + b_{i,x}$.

**Lemma A.2.** *Under the conditions of Theorem 3.1, with probability 1, the total number of distinct possible tuples $(W_{1,x}, b_{1,x}, \ldots, W_{d,x}, b_{d,x})$ satisfies*

$$|\{(W_{1,x}, b_{1,x}, \ldots, W_{d,x}, b_{d,x}) : x \in \mathbb{R}^k\}| \leq 10^{d^2}(n_1 \ldots n_d)^{d(k+1)}.$$

*Proof.* Let $S = \mathbb{R}^{k+1}$, which contains $(x, 1)$. Then the result of (Hand and Voroninski, 2017, Lemma 15) applied to the vector space $S$ and to $\tilde{W}_{1,x} = (W_{1,x} \ b_{1,x})$ yields

$$|\{(W_{1,x}, b_{1,x} : x \in \mathbb{R}^k)\}| \leq 10 n_1^{k+1}.$$

344 Each distinct $(W_{1,x}, b_{1,x})$ defines an affine linear space of dimension $k$ which contains the first layer
345 output $x_1$, and hence a subspace $S$ of dimension $k + 1$ which contains $(x_1, 1)$. Applying (Hand and
346 Voroninski, 2017, Lemma 15) to each such $S$ and $\tilde{W}_{2,x}$ yields

$$|\{(W_{2,x}, b_{2,x} : x \in \mathbb{R}^k)\}| \leq 10 n_1^{k+1} \cdot 10 n_2^{k+1}.$$

347 Proceeding inductively,

$$|\{(W_{i,x}, b_{i,x} : x \in \mathbb{R}^k)\}| \leq 10^i (n_1 \ldots n_i)^{k+1},$$

348 which is analogous to (Hand and Voroninski, 2017, Lemma 16) in our setting with biases $b_1, \ldots, b_d$.
349 The result follows from taking the product over $i = 1, \ldots, d$. $\square$

350 **Lemma A.3.** *Let $A \in \mathbb{R}^{m \times n}$ have i.i.d. $\mathcal{N}(0, 1/m)$ entries. Fix $\varepsilon > 0$, let $k < n$, and let*
351 *$V = \bigcup_{i=1}^M V_i$ and $W = \bigcup_{j=1}^N W_j$ where $V_i$ and $W_j$ are subspaces of dimension at most $k$. Then*
352 *with probability at least $1 - MN(c/\varepsilon)^{2k} e^{-c' \varepsilon m}$, for all $x \in V$ and $y \in W$ we have*

$$|x^\top A^\top A y - x^\top y| \leq \varepsilon \|x\| \|y\|.$$

353 *Proof.* See (Hand and Voroninski, 2017, Lemma 14). $\square$

354 Using these results, we analyze the gradient and critical points of $f(x)$. Note that with the above
355 definitions,

$$G(x) = V(W_{d,x} \ldots (W_{1,x} x + b_{1,x}) \ldots + b_{d,x})$$

$$= V \left( \prod_{i=d}^1 W_{i,x} \right) x + V \sum_{j=1}^d \left( \prod_{i=d}^{j+1} W_{i,x} \right) b_{j,x}.$$

356 The function $G(x)$ is piecewise linear in $x$, so $f(x)$ is piecewise quadratic. If $f(x)$ is differentiable
357 at $x$, then the gradient of $f$ can be written as

$$\nabla f(x) = \left( \prod_{i=1}^d W_{i,x}^\top \right) V^\top A^\top \left( AV \left( \prod_{i=d}^1 W_{i,x} \right) x + AV \sum_{j=1}^d \left( \prod_{i=d}^{j+1} W_{i,x} \right) b_{j,x} - Ay \right).$$

358 **Lemma A.4.** *Define*

$$g_x = 2^{-d} x - \left( \prod_{i=1}^d W_{i,x}^\top \right) V^\top y$$

359 *Under the conditions of Theorem 3.1, we have with probability $1 - C(e^{-c\varepsilon m} + e^{-c\varepsilon n} +$*
360 *$\sum_i n_i e^{-c\varepsilon^2 n_{i-1}})$ that at every $x \in \mathbb{R}^k$ where $f$ is differentiable,*

$$\|\nabla f(x) - g_x\| \leq C' \varepsilon (1 + \|x\| + \|y\|)$$

361 *Proof.* By Lemma A.2, for fixed $\theta = (V, W_1, b_1, \ldots, W_d, b_d)$, the range $\{V \prod_{i=d}^1 W_{i,x} x' : x, x' \in$
362 $\mathbb{R}^k\}$ belongs to a union of at most $C(n_1 \ldots n_d)^{d(k+1)}$ subspaces of dimension $k$. For some $C', c > 0$,
363 under the condition $m \geq C' k (\varepsilon^{-1} \log \varepsilon^{-1}) \log(n_1 \ldots n_d)$, we have

$$C^2 (n_1 \ldots n_d)^{2d(k+1)} (c/\varepsilon)^{2k} e^{-c' \varepsilon m} \leq e^{-c\varepsilon m}.$$

364 Then for $A \in \mathbb{R}^{m \times n}$ with i.i.d. $\mathcal{N}(0, 1/m)$ entries, applying Lemma A.3 conditional on $\theta$, and then
365 A.1(a) to bound $\|W_{i,x}\|$ and $\|V\|$, we get

$$\left\| \left( \prod_{i=1}^d W_{i,x}^\top \right) V^\top (A^\top A - I) V \left( \prod_{i=d}^1 W_{i,x} \right) x \right\| \leq C\varepsilon \|x\|.$$

366 For $A = I$, this bound is trivial. The given conditions imply also

$$n \geq n_d \geq C' k (\varepsilon^{-1} \log \varepsilon^{-1}) \log(n_1 \ldots n_d),$$

367   so applying the same argument with $V$ in place of $A$ yields

$$\left\|\left(\prod_{i=1}^{d} W_{i,x}^{\top}\right)(V^{\top}V - I)\left(\prod_{i=d}^{1} W_{i,x}\right)x\right\| \le C\varepsilon\|x\|.$$

368   Next, applying Lemma A.1(a–b) yields, for each $j = d, d-1, \ldots, 2, 1$,

$$\left\|\left(\prod_{i=1}^{j-1} W_{i,x}^{\top}\right)(W_{j,x}^{\top}W_{j,x} - I/2)\left(\prod_{i=j-1}^{1} W_{i,x}\right)x\right\| \le C\varepsilon\|x\|.$$

369   Combining these results, we get for the first term of $\nabla f(x)$ that

$$\left\|\left(\prod_{i=1}^{d} W_{i,x}^{\top}\right)V^{\top}A^{\top}AV\left(\prod_{i=d}^{1} W_{i,x}\right)x - 2^{-d}x\right\| \le C\varepsilon\|x\|. \qquad (3)$$

370   This holds with probability at least $1 - e^{-c\varepsilon m} - e^{-c\varepsilon n} - C\sum_i n_i e^{-cn_{i-1}}$.

371   The second term is controlled similarly: Lemma A.2 implies that for fixed parameters $\theta$, the set
372   $\{V \prod_{i=d}^{j+1} W_{i,x} b_{j,x} : x \in \mathbb{R}^k, j \in [d]\}$ is comprised of at most one of $C(n_1 \ldots n_d)^{d(k+1)}$ distinct
373   vectors (which belong to subspaces of dimension 1.) Then applying Lemma A.3 twice to $A$ and $V$ as
374   above, and using also $\|b_{j,x}\| \le 2$ from Lemma A.1(a),

$$\left\|\left(\prod_{i=1}^{d} W_{i,x}^{\top}\right)(V^{\top}A^{\top}AV - I)\left(\prod_{i=d}^{j+1} W_{i,x}\right)b_{j,x}\right\| \le C\varepsilon.$$

375   Applying Lemma A.1(a–b) iteratively as above, we get

$$\left\|\left(\prod_{i=1}^{j} W_{i,x}^{\top}\right)\left[\left(\prod_{i=j+1}^{d} W_{i,x}^{\top}\right)\left(\prod_{i=d}^{j+1} W_{i,x}\right) - 2^{-(d-j)}I\right]b_{j,x}\right\| \le C\varepsilon.$$

376   Finally, Lemma A.1(a) and (c) yield

$$\left\|\left(\prod_{i=1}^{j} W_{i,x}^{\top}\right)b_{j,x}\right\| \le C\varepsilon.$$

377   Combining these, we have for the second term of $\nabla f(x)$ that

$$\left\|\sum_{j=1}^{d}\left(\prod_{i=1}^{d} W_{i,x}^{\top}\right)V^{\top}A^{\top}AV\left(\prod_{i=d}^{j+1} W_{i,x}\right)b_{j,x}\right\| \le C\varepsilon \qquad (4)$$

378   also with probability $1 - e^{-c\varepsilon m} - e^{-c\varepsilon n} - C\sum_i n_i e^{-c\varepsilon^2 n_{i-1}}$.

379   Finally, for the last term of $\nabla f(x)$, if $A \ne I$ then we may apply Lemma A.3 again to get

$$\left\|\left(\prod_{i=1}^{d} W_{i,x}^{\top}\right)V^{\top}(A^{\top}A - I)y\right\| \le C\varepsilon\|y\| \qquad (5)$$

380   with probability $1 - e^{-c\varepsilon m}$. Combining (3), (4), and (5) concludes the proof.   $\square$

381   We now bound the second term of $g_x$.

382   **Lemma A.5.** *Under the conditions of Theorem 3.1, with probability $1 - Cn_d e^{-c\varepsilon^4 n_{d-1}}$, for every*
383   $v \in \mathbb{R}^{n_{d-1}}$

$$\left\|W_{d,+,v}^{\top}V^{\top}y\right\| \le C\varepsilon\|y\|.$$

384   *Proof.* Note that $V^{\top}y \in \mathbb{R}^{n_d}$ has i.i.d. $\mathcal{N}(0, \|y\|^2/n)$ entries. Then conditional on $W_d$, for each
385   fixed $v \in \mathbb{R}^{n_{d-1}}$,

$$u(v) \equiv W_{d,+,v}^{\top}V^{\top}y \sim \mathcal{N}(0, \Sigma)$$

386 where
$$\Sigma = (\|y\|^2/n) \cdot W_{d,+,v}^\top W_{d,+,v} \in \mathbb{R}^{n_{d-1} \times n_{d-1}}.$$

387 On the event that Lemma A.1(b) holds, we have $\|\Sigma\| \leq \|y\|^2/n$ and hence $\|u(v)\|^2 \leq tn_{d-1}\|y\|^2/n$
388 with probability $1 - e^{cn_{d-1}t}$ for large $t$, by a $\chi^2$ tail-bound. Noting that $n \geq n_d \gg \varepsilon^{-2}n_{d-1}$ and
389 applying this bound for $t = \varepsilon^2 n/n_{d-1}$, we get $\|u(v)\| \leq \varepsilon\|y\|$ with probability $1 - e^{-c\varepsilon^2 n}$.

390 We use a covering net argument to take a union bound over $v$: Let $N$ be an $\varepsilon^2$-net of the $n_{d-1}$-sphere,
391 of cardinality $|N| \leq (3/\varepsilon^2)^{n_{d-1}}$. The above holds uniformly over $v \in N$ with probability $1 - e^{c'\varepsilon^2 n}$,
392 because $n \geq n_d \gg n_{d-1} \cdot \varepsilon^{-2}\log\varepsilon^{-1}$. For any $v'$ on the sphere and $v \in N$ with $\|v - v'\| < \varepsilon^2$, the
393 angle $\theta$ between $v$ and $v'$ is at most $C\varepsilon^2$. We have
$$\|u(v) - u(v')\| \leq \left\| W_{d,+,v}^\top - W_{d,+,v'}^\top \right\| \cdot \|V^\top y\|.$$

394 Suppose now that Lemma A.1(b) holds for $W_d$ with the constant $\varepsilon^2$: This occurs with probability
395 $1 - 8n_d e^{-c\varepsilon^4 n_{d-1}}$. Approximating each of the four terms in
$$\left(W_{d,+,v}^\top - W_{d,+,v'}^\top\right)\left(W_{d,+,v} - W_{d,+,v'}\right)$$

396 by $I/2$ on this event, we get
$$\left\| W_{d,+,v}^\top - W_{d,+,v'}^\top \right\|^2 = \left\| \left(W_{d,+,v}^\top - W_{d,+,v'}^\top\right)\left(W_{d,+,v} - W_{d,+,v'}\right) \right\| \leq C'(\varepsilon^2 + \theta) \leq C\varepsilon^2.$$

397 Thus on this event, $\|u(v) - u(v')\| \leq C\varepsilon\|V^\top y\|$. By a $\chi^2$ tail-bound, with probability $1 - e^{-cn_d}$
398 we have $\|V^\top y\|^2 \leq 2\|y\|^2 n_d/n \leq 2\|y\|^2$ and hence $\|u(v) - u(v')\| \leq C\varepsilon\|y\|$. $\square$

399 *Proof of Theorem 3.1.* Combining Lemmas A.4, A.5, and A.1(a), with the stated probability,
$$\|\nabla f(x) - 2^{-d}x\| \leq C\varepsilon(1 + \|x\| + \|y\|)$$

400 for every $x \in \mathbb{R}^k$. Since $G$ is piecewise linear, the directional derivative $D_v f(x)$ always exists at any
401 $x \in \mathbb{R}^k$ for any unit vector $v \in \mathbb{R}^k$, even for $x$ where $f$ is non-differentiable. Set $\tilde{x} = x/\|x\|$. For any
402 fixed $x$, there exists a sequence $\{x_n\}$ which converges to $x$ and where $f$ is differentiable, such that
$$D_{-\tilde{x}}f(x) = \lim_{n\to\infty} -\tilde{x}^\top \nabla f(x_n)$$

403 Since
$$-\tilde{x}^\top \nabla f(x_n) = -2^{-d}\tilde{x}^\top x_n + \tilde{x}^\top(2^{-d}x_n - \nabla f(x_n)) \leq -2^{-d}\tilde{x}^\top x_n + C\varepsilon(1 + \|x_n\| + \|y\|),$$

404 we get
$$D_{-\tilde{x}}f(x) \leq \liminf_{n\to\infty} \left[ -2^{-d}\tilde{x}^\top x_n + C\varepsilon(1 + \|x_n\| + \|y\|) \right]$$
$$= -2^{-d}\|x\| + C\varepsilon(1 + \|x\| + \|y\|).$$

405 For $\varepsilon > 0$ sufficiently small and $C' > 0$ sufficiently large, this implies $D_{-\tilde{x}}f(x) < 0$ whenever
406 $\|x\| \geq C'\varepsilon(1 + \|y\|)$. $\square$

# B  Comment on Projected-Gradient Surfing

408 The projected-gradient surfing algorithm performs an exhaustive search over pieces $P_g \in$
409 $\mathcal{P}(x_{t-1}, \theta(\delta t), \tau)$. The number of such pieces is at most $1 + 2^{|S(x_{t-1}, \theta(\delta t), \tau)|}$, where we recall
410 that
$$S(x, \theta, \tau) = \{(i,j) : |w_{i,j}^\top x_{i-1} + b_{i,j}| \leq \tau\}$$

411 is the collection of layers and rows where the sign could change during the next step.

412 We reason heuristically that if $\theta \equiv \theta(\delta t)$ is "generic", then for sufficiently small $\tau$, we should have
413 $|S(x, \theta, \tau)| \leq dk$ for all $s \in [0, S]$ and $x \in \mathbb{R}^k$, so that this search is tractable for small $k$. Indeed,
414 for fixed $W_1, b_1, \ldots, W_i, b_i$, the set of possible outputs $\{x_i : x \in \mathbb{R}^k\}$ at the $i^{\text{th}}$ layer is a finite
415 union of affine linear spaces of dimension $k$. For generic $W_{i+1}$ and $b_{i+1}$, and every $J \subset [n_i]$ where
416 $|J| = k + 1$, each such space has empty intersection with the affine linear space
$$\{z \in \mathbb{R}^{n_i} : w_{i+1,j}^\top z + b_{i+1,j} = 0 \text{ for all } j \in J\}$$
417 of dimension $n_i - k - 1$. Thus
$$\sup_{x \in \mathbb{R}^k} |\{j \in [n_i] : w_{i+1,j}^\top x_i + b_{i+1,j} = 0\}| \leq k,$$

418 so $\sup_{x \in \mathbb{R}^k} |S(x, \theta, 0)| \leq dk$ for $\tau = 0$. Then we expect this to hold also for some small $\tau > 0$.

## C Additional Simulations

Here we give additional plots for experiments comparing surfing over a sequence of networks during training to gradient descent over the final trained network. As described in the main text, we consider the problem of minimizing the objective $f(x) = \frac{1}{2}\|G(x) - G(x_*)\|^2$, that is, recovering the image generated from a trained network $G(x) = G_{\theta_T}(x)$ with input $x_*$. We run surfing by taking a sequence of parameters $\theta_0, \theta_1, ..., \theta_T$, where $\theta_0$ are the initial random parameters and the intermediate $\theta_t$'s are taken every 40 training steps. In order to improve convergence speed we use Adam (Kingma and Ba, 2014) to carry out gradient descent in each step in surfing. We also use Adam when optimizing over the just the final network. We apply surfing and regular Adam for 300 trials, where in each trial a randomly generated $x_*$ and initial point $x_{init}$ is chosen. Figure 5 shows the distribution of the distance between the computed solution $\widehat{x}_T$ and the truth $x_*$ for VAE, WGAN and WGAN-GP, using surfing (red) and regular gradient descent with Adam (blue), over three different input dimensions $k$.

Figure 5: Distribution of the distance between solution $\widehat{x}_T$ and the truth $x_*$ for VAE, WGAN and WGAN-GP, using surfing (red) and regular gradient descent with Adam (blue) over three different input dimensions $k$.