[Reviews · NeurIPS 2019]

Reviewer 1



Originiality and Quality: 1. The work is novel and the algorithm proposed for empirical minimization is novel. However, I have some issues with the motivation and experiments, listed under weaknesses below. 2. Theorem 3.1 is interesting in itself, because it applies to vectors which are not in the range of the generative model. Clarity: The paper is well written and ideas clearly expressed. I believe that others can reproduce the algorithm described. I only have a problem with the way the set $S(x,theta, tau)$ is defined in line 177, since the authors do not require the signs to strictly differ on this set. Significance: I think other researchers can build on Theorem 3.1. The conditions proposed for Theorem 3.3 are novel and could be used for future results. Weaknesses: 1. I am not completely convinced by the experimental strengths of this approach. To run the proposed algorithm, the authors need to run a descent procedure for 40 different networks from the training phase. In contrast, you could simply run vanilla Adam on the final network with 40 random initial points, and one of these restarts would reach the global minimum. It is not important that EACH initialization reach the global minimum, as long as AT LEAST one initialization reaches the global minimum. 2. While Theorem 3.3 is interesting, it does not directly influence the experiments because the authors never perform the search operation in line 3 of algorithm 2. Because of this, it provides a proof of correctness for an algorithm that is quite different from the algorithm used in practice. Although Algorithm 2 and the empirical algorithm are similar in spirit, lines 1 and 3 in algorithm 2 are crucial for proof of correctness. Clarifications: 1. For the case where $y= G(z) + noise$, where noise has sufficiently low energy, you would expect a local minimum close to $z$. Would this not contradict the result of Theorem 3.1? ---Edit after author response--- Thank you for your response. After reading your rebuttal and other reviews, I have updated my score to a 8. I think Table in the rebuttal and Theorem 3.1 are solid contributions. Regarding my criticism of the definition of S(x,tau,theta)- I only meant that defining the complement of this set may make things clearer, since you only seem to work with its complement later on (this did not influence my score).

Reviewer 2



In this work, the authors study the idea of surfing, the optimization of a non-smooth model by sequentially optimizing a smoothly varying sequence of models that interpolate between something very smooth and the desired non-smooth model. This sequence of models can be obtained, for example, by snapshots of the model during the training process. The authors provide empirical studies that show that some models which can not directly be optimized, can be successfully optimized by this process. This has significant ramifications for solving inverse problems using generative models, as optimization over GANs has been shows to succeed with quite few measurements. The authors present a clearly defined probability model for networks at initialization, for which they present analysis. The authors then show that if a sequence of networks is given by a discretization of a continuous flow (in which the global optimizer moves in a Lipschitz way), then a projected gradient descent procedure successfully optimizes the models. Limitations of the theoretical analysis are that the complexity of the algorithm depends on the number of pieces of the piecewise linear model G that are optimized within each step. A heuristic is provided for this, but ideally a full proof would be established. Despite this limitation, the paper does offer an insightful observation and analysis to the field. The paper is clearly written, with well chosen figures and easy to follow text. I think the paper could be improved by having a discussion about the tradeoffs involved. For example, the sequential optimization procedure sounds like it could be quite expensive. One would need to store a full sequence of models, and reconstruction time may be quite slow. I would enjoy hearing informed thoughts on the ramifications for all this additional computation involved.

Reviewer 3



The paper is in general well written. The results are built upon the result in (Hand and Voroninski, 2017), and (Bora et al. 2017). From my understanding, the first theorem is mainly built on (Hand and Voroninski, 2017), and the second theorem is mainly built on (Bora et al. 2017). For the second theorem, the result implies the deeper the network is, the smaller the delta should be. It would be better to discuss how tight is the analysis, and whether this dependency is necessary in practice. ==== After rebuttal I have carefully read the author's response and other reviewer's feedback. I have less concern about the usefulness of the theoretic results and the gap between the theory and the algorithm. I have changed my score from 6 to 7. I would expect a higher score with much stronger experiments, although proving the idea of the algorithm works itself is quite interesting.

[Author Response · NeurIPS 2019]

| | Model | DCGAN | | | WGAN | | | WGAN-GP | | |
|---|---|---|---|---|---|---|---|---|---|---|
| | Input dimension | 5 | 10 | 20 | 5 | 10 | 20 | 5 | 10 | 20 |
| % successful | Regular Adam | 48.3 | 68.7 | 80.0 | 56.0 | 84.3 | 90.3 | 47.0 | 64.7 | 64.7 |
| | Surfing | 78.3 | 98.7 | 96.3 | 81.7 | 97.3 | 99.3 | 83.7 | 95.7 | 97.3 |
| # iterations | Regular Adam | 618 | 4560 | 18937 | 464 | 1227 | 3702 | 463 | 1915 | 15445 |
| | Surfing | 741 | 6514 | 33294 | 547 | 1450 | 4986 | 564 | 2394 | 25991 |

Table 1: Surfing compared against direct gradient descent over the final trained network. Shown are percentages of "successful" solutions $\hat{x}_T$ satisfying $\|\hat{x}_T - x_*\| < 0.01$, and 75th-percentiles of the total number of gradient descent steps used (across all networks $G_0, \ldots, G_T$ for surfing) until $\|\hat{x}_T - x_*\| < 0.01$ was reached.

We thank the reviewers for carefully reading our paper and providing insightful and constructive comments. We will respond to each of the concerns that were raised.

*Reviewers 1 and 2 both comment on the computational cost of the procedure, compared with running vanilla Adam with multiple random initial points.* We thank the reviewers for raising this important point, which led us to further explore the computational cost of surfing. In fact, surfing can be performed such that its runtime is close to that of a *single* initialization of vanilla Adam—the reason is that for the intermediate networks, gradient descent (GD) does not need to be run until full convergence; the number of GD steps can be quite small and surfing will still succeed.

The updated Table 1 illustrates this: Briefly, we re-ran both vanilla Adam and surfing on the DCGAN, WGAN, and WGAN-GP examples, using the same step size in both methods. We recorded the 75th-percentile of the number of GD steps $N$ needed in vanilla Adam to achieve $\|\hat{x}_T - x_*\| < 0.01$. We then constrained surfing to use $N$ total iterations across networks $G_0, \ldots, G_{99}$, followed by GD until convergence for the final trained network $G_{100}$. The $N$ steps in surfing were split across networks $G_0, \ldots, G_{99}$ proportional to a common deterministic schedule, which alloted more steps to the earlier networks $G_t$ where the landscape changes more rapidly, and fewer steps to later networks where this landscape stabilizes. Shown are the success rates and the 75th-percentiles of the total number of GD iterations for both methods. We see that surfing still has a much higher success rate, at a comparable computational cost to a single initialization for vanilla Adam. We will update Table 1 of the original manuscript to display this new comparison.

*R1: I only have a problem with the way the set $S(x, \theta, \tau)$ is defined in line 177, since the authors do not require the signs to strictly differ on this set.* $S(x, \theta, \tau)$ is just the set of neurons that are close to zero before ReLU thresholding. These are the neurons for which the signs could change after a small change of the network input $x$.

*R1: Although Algorithm 2 and the empirical algorithm are similar in spirit, lines 1 and 3 in algorithm 2 are crucial for proof of correctness.* Theorem 2 mainly illustrates that the procedure can be formalized, although in its current form the projected gradient algorithm is not easily implemented.

*R1: For the case where $y = G(z) + noise$, where noise has sufficiently low energy, you would expect a local minimum close to $z$. Would this not contradict the result of Theorem 3.1?* This case is not covered by Theorem 3.1, because $y$ is then correlated with the network parameters. Please see our comment starting on line 157.

*R2: I find the paper quite interesting already. To make it even more interesting would involve having a complete theoretical argument establishing the time complexity without the current heuristic.* We agree that a full theoretical analysis would be preferred. Ultimately we think that something between the simple surfing and projected gradient surfing methods will be more attractive in both theory and practice.

*R3: From my understanding, the first theorem is mainly built on (Hand and Voroninski, 2017), and the second theorem is mainly built on (Bora et al.)* Our analysis builds primarily on Hand and Voroninski. The type of result in Bora et al. is different, and pertains to properties of near-global minimizers rather than computational procedures for finding them.

*R3: For the second theorem, the result implies the deeper the network is, the smaller the delta should be. It would be better to discuss how tight is the analysis, and whether this dependency is necessary in practice.* The dependence of $\delta$ on network depth comes from upper-bounding the Lipschitz constant of the network $G(x)$ by $\prod_{i=1}^{d} \|W_i\|$. We do expect the true Lipschitz constant to increase with network depth in practice. The upper-bound is likely not tight, but it may be difficult to theoretically improve. The same type of bound was used in Szegedy et al. (2014); Virmaux and Scaman (2018) which discussed this question in more detail—we will add a discussion of this point to the manuscript.

# References

Szegedy, C., Zaremba, W., Sutskever, I., Bruna, J., Erhan, D., Goodfellow, I., and Fergus, R. (2014). Intriguing properties of neural networks. In *International Conference on Learning Representations*.

Virmaux, A. and Scaman, K. (2018). Lipschitz regularity of deep neural networks: Analysis and efficient estimation. In *Advances in Neural Information Processing Systems*, pages 3835–3844.


[Meta-Review · NeurIPS 2019]

The paper proposes a new method for provably fitting deep generative models to observations, a highly non-convex optimization problem. Instead of trying to find the latent code that explains the measurements directly, as proposed by Bora et al. this paper starts with a different deep generative model that has random weights, for which Hand et al. showed that gradient descent provably works. Then they incrementally modify the weights of the generator to approach the true generator while using the previous optimum as a starting point. This sequence of models can be snapshots of the model during the training process. The main result is a theory that shows that a warm-started non convex optimization in expansive Gaussian networks yields successful recovery. The proposed method, Surfing, can be seen as a variation of multiple random restarts. Instead of randomly restarting, the network is slightly modified and the previous starting point is used. Reviewers 1 and 2 brought up concerns about the running time of the proposed method as opposed to random restarts but the rebuttal sufficiently addressed this. This paper makes important contributions for nonconvex optimization for deep learning.